# Correlation of Kinematics and Kinetics of Changing Sagittal Plane Body Position during Landing and the Risk of Non-Contact Anterior Cruciate Ligament Injury

**Mahgolzahra Kamari** [1],*,† , **Randeep Rakwal** [2],† , **Takuya Yoshida** [2], **Satoru Tanigawa** [2] and **Seita Kuki** [3]

1. Graduate School of Comprehensive Human Sciences, Tsukuba International Academy for Sport Studies (TIAS), Tsukuba 305-8574, Japan
2. Faculty of Health and Sport Sciences, University of Tsukuba, Tsukuba 305-8574, Japan; plantproteomics@gmail.com (R.R.); yoshida.takuya.gm@u.tsukuba.ac.jp (T.Y.); tanigawa.satoru.gb@u.tsukuba.ac.jp (S.T.)
3. Faculty of Human Sciences, University of Economics, Higashiyodogawa-ku, Osaka 533-8533, Japan; kukiseita0607@gmail.com
* Correspondence: zahra.kamari67@gmail.com
† Joint first authors.

**Abstract:** Anterior cruciate ligament (ACL) injury is one of the most common knee injuries that negatively affect athletes' future performance and return to play. The purpose of this study was to examine the correlation of kinematics and kinetics of changing sagittal plane body position during landing and the risk of non-contact ACL injury. Seven university female (age $19.57 \pm 0.79$ y, height $164.21 \pm 8.11$ m, weight $60.43 \pm 5.99$ kg) athletes playing soccer and handball, and with $\geq$ two years of training volunteered for this research. Three trunk positions: Lean Forward Landing (LFL), Self-selected Landing (SSL), and Upright Landing (URL)—via double/single-leg landing—were captured by a high-speed VICON motion capture system. A $3 \times 2$ two-way within-subjects ANOVA and Multiple Bonferroni corrected pairwise were used to test for condition (LFL, SSL, URL) and task (single/double-leg) effects ($p \leq 0.05$). The findings indicated that landing with a deeper knee flexion angle (LFL) would lead to smaller impact forces when compared to upright landing.

**Keywords:** ACL injuries; landing trunk positions; non-contact; sagittal plane; women athlete

## 1. Introduction

Surveying the research literature, sports-related websites and the news confirm that knee injuries are common among athletes and the active population. It can be considered to be one of the biggest contributors among all sport injuries, accounting for around 15–50 percent [1]. Further, women have a more pronounced risk of sustaining a knee injury compared to males [1,2]. Research efforts over the years have found that a majority of the injuries among female athletes were found particularly in lower extremities such as the ankle and the knee [3,4]. The Anterior Cruciate Ligament (ACL) is the site of the most serious and common knee injuries, and has been studied in many different ways, especially over the past few years, due to the increased number of participating athletes in research studies [5]. The knee joint is the common site of injury only second to the ankle [6], and in terms of severity, the rupture of the ACL can often cause critical loss of maximum performance for athletes [6–8].

Previous studies [9–11] have investigated a range of biomechanical motions that are concerned with the risks of ACL injuries. In addition to the lower limb adaptations at the ankle and knee, the upper body trunk motion has been considered to contribute to impact dissipation of the ACL. Studies have shown that the upright trunk position [4,8] while landing [12] and inability to control the frontal plane trunk position [8,12] are often associated with ACL injuries. Therefore, it is important to assess how trunk control affects

lower extremity biomechanical parameters that may be associated with reducing the ACL injuries. Trunk stability is related to the ability of the hip to control the trunk in response to forces generated from distal body segments and unexpected perturbations. Greater trunk flexion during landing may contribute to lower ACL loading and associated injury risk if this joint motion was effective in reducing ground reaction forces (GRF) [13,14]. Therefore, the purpose of this study was to determine the effect of trunk posture on key biomechanical variables that indicate ACL injury risk such as drop landing forces. It was hypothesized that the most ACL-protective biomechanical parameters would be associated with leaning forward landings (LFL) and the most harmful with upright landings (URL).

Single- and double-leg stop-jump techniques are frequently executed in the competitive sports such as basketball, handball and soccer. Yu et al. (2006) [15] indicated that the landing phase of stop-jump tasks presents a significant risk of injury to the lower extremities in general, particularly to the ACL. A number of reports have shown that most sports-related ACL injuries occur during non-contact situations that are characterized by landing, rapid deceleration, and sudden changes of direction [16,17]. In a single-leg drop landing task, larger GRF, an increased knee valgus, and decreased knee flexion were identified at initial contact compared to double-leg drop landing [18,19]. This incidence rate for non-contact ACL injuries is two to eight times higher for female athletes compared to males in the same sport [20,21]. It is, therefore, interesting to investigate types of trunk motion that can improve landing mechanics in both single-leg and double-leg landing.

This current research is focused on: (1) evaluating the effect of trunk body position on leaning forward landing (LFL), self-selected landing (SSL) and upright landing (URL) during a single/double-leg drop landing for ACL injuries risk reduction in university female players, and (2) investigating the risk of non-contact ACL injury in maximum strength jump-landing during landing position. An in-depth study on trunk body position during drop landing can possibly have implications to reduce the ACL injury risk among elite female players.

## 2. Materials and Methods

A quantitative laboratory study with sagittal plane knee, hip, and trunk kinematics and kinetics in female athletes was performed.

### 2.1. Participants

Seven female university-level athletes (age $19.57 \pm 0.79$ y, height $164.21 \pm 8.11$ m, weight $60.43 \pm 5.99$ kg) playing soccer and handball, with $\geq 2$ years of training volunteered in this investigation. Every subject was provided with an explanation about the experimental protocol. Each participant signed the informed consent prior to participation in the research, which was granted and mandated by the University of Tsukuba Research Ethics Committee. Inclusion criteria for participation were as follows: female, aged 18–25 and an active player on a university sports team involved in regular physical activity at least three days per week with each session longer than 45 min. Prospective participants were excluded if any of the following exclusion criteria were identified: known history of ACL injury or reconstruction; known history of a surgical procedure involving the lower extremity; previous lower extremity musculoskeletal injury in the last six weeks that required medical attention.

### 2.2. Protocols

Drop-jumps were conducted to characterize landing mechanics in biomechanical testing environments including the two force plates (Kistler 9287C, 60 cm × 40 cm) to collect the GRFs at 1000 Hz and 10 high-speed motion capture cameras (Vicon motion analyzer MX, type of T20 and T20S) were used to sample whole body kinematics at 250 Hz. For drop-jumps, a box with a height of 30 cm was placed 10 cm away from the force plate, and prior to their jump participants would stand 1 cm from the edge of the box. This box

was placed directly between two adjacent force platforms, where the box was positioned 15 cm away from the edge of the platforms (Figure 1).

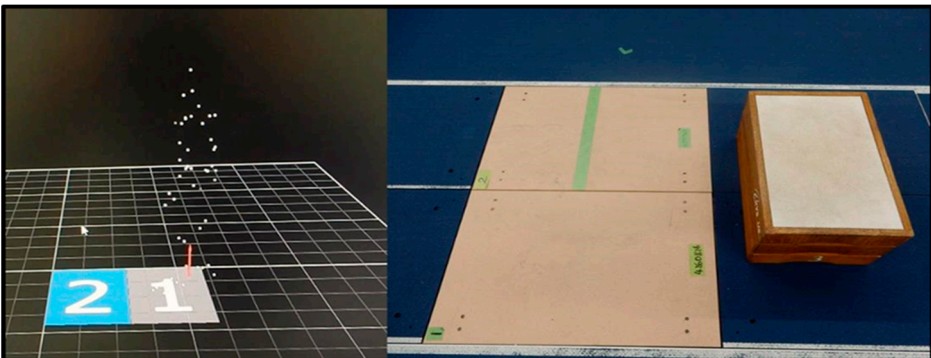

**Figure 1.** Global coordinate system and force plate orientation (Kistler 9287C).

One hour before the experiment was initiated examiners set the VICON cameras, force plate, retroreflectors, computers, and recorders setting and prepared the clothes for the subjects. Every subject was provided general information about the experimental protocol.

Appropriate clothing consisted of own running shoes, tight-fitting top and high-cut running shorts. The warm-up consisted of a 10 min jog followed by light stretching. Participants completed a 5 min warm-up and 5 min self-directed stretching, prior to data collection. General anthropometric measures (height, and weight) were taken for each participant by the same researcher. As a specific warm-up, all the participants practiced five landing tasks to become accustomed to the movements before data collection.

Forty-seven retro-reflective markers were applied bilaterally on specific body landmarks over the right hand, right wrist lateral side, right wrist medial side, right elbow lateral side, right elbow medial side, right shoulder front side, right shoulder back side, right acromial, left hand, left wrist lateral side, left wrist medial side, left elbow lateral side, left elbow medial side, left shoulder front side, left shoulder back side, left acromial, right toe, right ball lateral side, right ball medial side, right heel, right ankle lateral side, right ankle medial side, right knee lateral side, right knee medial side, right trochanterion, left toe, left ball lateral side, left ball medial side, left heel, left ankle lateral side, left ankle medial side, left knee lateral side, left knee medial side, left trochanterion, top of head, right ear, left ear, suprasternal front, suprasternal back, right rib, left rib, xiphoid, xiphoid back, right anterior-superior iliac spine, left anterior-superior iliac spine, right posterior-superior iliac spine, and left posterior-superior iliac spine (Figure 2).

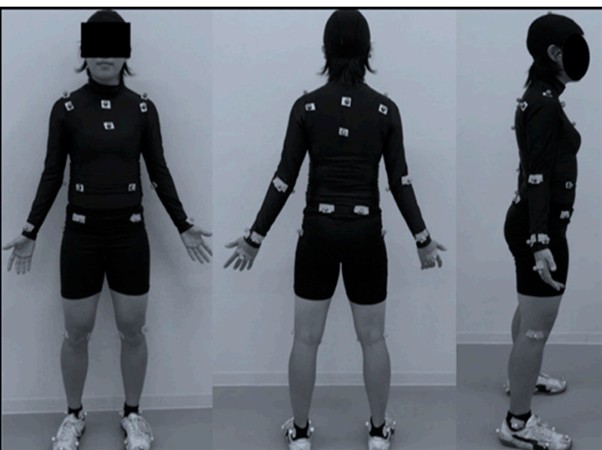

**Figure 2.** A total of 47 reflective markers were placed on each subject.

Seven participants were required to partake in two different landing techniques and were permitted three practice trials for each technique. Five successful trials, verified by video, were collected for each trunk posture. If participants did not place the dominant foot (which was determined at the beginning of experiment by asking each participant; all of them were aware of their dominant foot) on the force plate, or lost balance and did not execute the task with the appropriate landing technique or did not perform the appropriate task based on the cue generated by the software, the trial was not successful and discarded from analysis. Participants were initially instructed to stand on the edge of the box with feet 35 cm apart. They stood on the step on their dominant leg with the knee of the other leg bent at approximately 90°, with neutral hip rotation, arms crossed/with their hands on their hips. Subjects for single-leg landing stood on the opposite foot with their hands on their hips and landed with the foot in the center of the force plate. Then, drop-landings were performed in three trunk positions (LFL, URL, and SSL) randomly. For double leg landings, participants were instructed to drop off the box, with both feet landing simultaneously and entirely on the force platforms (one platform per leg) to complete five trials as landed. After landing (single-leg/double-leg) on the force plate with the dominant leg, participants were required to hold this position for three seconds.

Upon the completion of the test, the reflective markers were removed. This research defined an action cycle as landing and keeping balance on the ground. The moment of foot initial contact to the ground is the time when force value is greater than zero in the first frame data that appears from the force platform. The action process of the single/double footstep and immediate stop stage is from the first touchdown of the foot to the knee flexion angle reaching the maximum. The biomechanical parameter extraction in two stages of this research included: (1) joint angle and angular velocity of the trunk, hip, knee and ankle during the initial touchdown of feet; (2) maximum flexion angle of the hip and knee; (3) the peak of GRF in the horizontal plane on single/double footstep and immediate stop stage and (4) the peak of GRF in the vertical plane on single/double footstep stage. Only successful drop landings (initial landing) of the trials were analyzed (Figure 3).

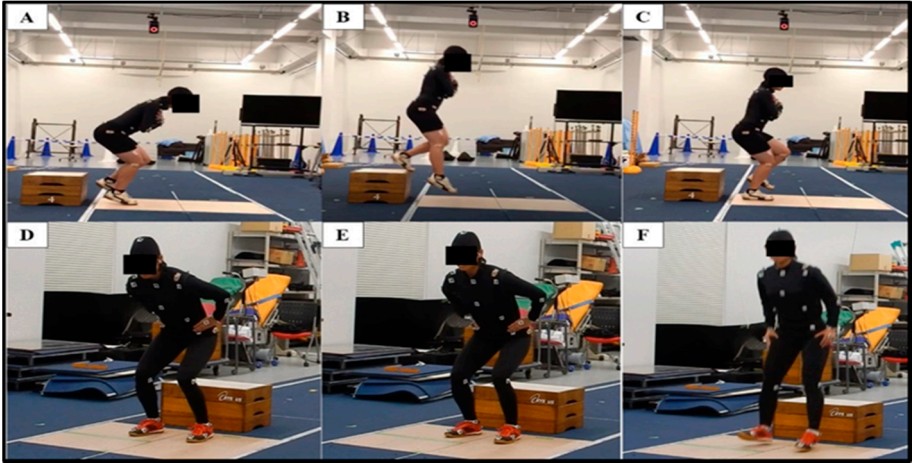

**Figure 3.** Sequence progression of the landing; (**A**) Single-leg LFL, (**B**) Single-leg SSL, (**C**) Single leg UPL, (**D**) Double-leg LFL, (**E**) Double-leg SSL, (**F**) Double-leg UPL.

*2.3. Parameters*

Kinematic and kinetics of joint angle and torque moment of the lower limb (hip, knee, and ankle) during three trunk positions (Lean forward, Self-selected, Upright landing) from single-leg and double-leg landings were completed.

2.3.1. Upright Landing (URL)

Initial contact during landing is on the heel with the body as upright as possible.

### 2.3.2. Lean Forward Landing (LFL)

Initial contact during landing is on the ball of the feet with the trunk leaning forward.

### 2.3.3. Self-Selected Landing (SSL)

Subject lands on the mid-foot with upright right trunk position. Participants were informed how to land self-selected prior to the start of the experiment.

### *2.4. Data Processing*

The kinetics and kinematic data were processed with a low-pass fourth-order zero-lag digital Butterworth filter at 40 and 12 Hz, respectively. The position and orientation data were synchronized with the GRF data. The joint angles were calculated as previously reported by Shultz and Schmitz [22]. To compare trunk inclination across the landing conditions, sagittal plane angles of the thorax and sacrum were calculated relative to the vertical axis of the global coordinate system and were defined as the thoracic and sacral inclination angles, respectively. The participants may have modified their sacral and thoracic inclination angles differently while landing due to different vertebral articulations in the lumbar spine. Therefore, both the sacral and thoracic inclination angles were examined to determine how the participants modified their trunk positions during single/double-leg landings. The thoracic and sacral inclination angles as well as the ankle dorsi/plantar flexion angles at initial foot contact were extracted to ensure that the participants successfully modified their landing styles across the three conditions. To test hypotheses of the current study, the peak vertical GRFs and peak plantar flexor moments as well as the times to these peaks were extracted. The peak knee extensor moment, sagittal plane hip and ankle moments, and knee flexion angles at the instant of peak knee extensor moment were subsequently extracted.

### *2.5. Statistical Methods*

Fifteen female subjects (university-level competitive athletes) volunteered to participate in this study; however, five did not meet study eligibility criteria due to the previous history of ACL injury. This left a total of 10 subjects who consented and participated in the study; however, three of them had to be excluded due to inadequate three-dimensional marker tracking during the landing phase of the landing trials. Landing biomechanics related to ACL injury were assessed during a double-leg/single-leg stop-landing maneuver, using a video-based motion analysis system. A repeated measure two-way ANOVA ($p \leq 0.05$) was used to test the within-subject differences of landing biomechanical characteristics between conditions (LFL, SSL, URL) and task (double and single leg). Regarding the ANOVA test, if the Mauchly's *p*-value was significant ($p \leq 0.05$), Greenhouse–Geisser correction results were taken from the tables, but, if the Mauchly's test *p*-value were found not significant ($p > 0.05$) sphericity-assumed results were used. Bonferroni design was used to prevent data from incorrectly appearing to be statistically significant.

## 3. Results

### *3.1. Demographic Results*

Table 1 shows the descriptive statistics of the three professional handball players and four professional soccer players who were eligible for complete inclusion in the study.

**Table 1.** Subject's demographic distribution (n = 7).

|  | **Min** | **Max** | **Mean** | **SD** |
|---|---|---|---|---|
| Age | 19 | 21 | 19.57 | 0.787 |
| Height (cm) | 153 | 177 | 164.21 | 8.113 |
| Weight (kg) | 53 | 68 | 60.43 | 5.987 |

### 3.2. Torque Analyses

This part will show the analyses of torque for the ankle (dorsiflexion), knee (flexion) and hip (flexion) in the three trunk positions while landing.

Table 2 shows the descriptive statistics and the results of ANOVA. *p*-values from ANOVA * denote statistically significant difference ($p \leq 0.05$).

**Table 2.** Descriptive statistics and results of 2-way repeated measure ANOVAs (Mean ± SD) comparisons of sagittal plan trunk angles (LFL, SSL and URL) and hip, knee and ankle flexion angles during double/single-leg landing.

| | LFL | | SSL | | URL | |
|---|---|---|---|---|---|---|
| | **Double Leg** | **Single Leg** | **Double Leg** | **Single Leg** | **Double Leg** | **Single Leg** |
| Hip | −49 ± 12.8 | −51.7 ± 22.3 | −43.3 ± 13.7 | * −63.9 ± 23.1 | * −44.2 ± 13.1 | * −59.2 ± 21.9 |
| Knee | −44.8 ± 13.6 | −61.2 ± 22 | −44.9 ± 12.7 | −64.9 ± 29.5 | −46.7 ± 16.6 | * −59.2 ± 26.1 |
| Ankle | * 46 ± 8 | * 48 ± 13 | 45 ± 11 | 40 ± 6 | 58 ± 30 | 66 ± 34 |

* Significant main effect at a level of less than 0.05. Negative directions indicate hip and knee flexion, positive directions indicates ankle dorsiflexion. LFL leaning forward landing, SSL self-selected landing, URL upright landing.

Table 2 shows that the ankle dorsiflexion was significant during single/double-leg landing in the LFL trunk position ($p \leq 0.05$).

The data in Table 2 also demonstrate significant difference levels of single-leg landing during URL for the knee and significant difference levels of URL position during single/double-leg landing for the hip. Moreover, result shows SSL was notably different for the hip during single-leg landing ($p \leq 0.05$).

### 3.3. Joint Angle Analyses

Table 3 shows the group means and standard deviations of the sagittal plane trunk angle (LFL, SSL, URL) during double leg and single leg landings among seven athletes (N). It was hypothesized that the dorsiflexion joint angle would be more significant for ACL injury in the URL during the landing phase. Further, based on estimated marginal means, the *p*-values from ANOVA; * denote statistically significant difference (16.7 ± 21.2). Moreover, results show the knee joint angle was significantly different during both landing phases in the URL trunk position ($p \leq 0.05$). *p*-value from ANOVA * denotes statistically significant differences for SSL and URL during single-leg landing (−81.9 ± 33.8 and −87.6 ± 50.7, respectively).

**Table 3.** Descriptive statistics and results of 2-way repeated measures ANOVA (Mean ± SD) comparisons of sagittal plan trunk angles (LFL, SSL and URL) and hip, knee and ankle flexion angles during double/single-leg landing.

| | LFL | | SSL | | URL | |
|---|---|---|---|---|---|---|
| | **Double Leg** | **Single Leg** | **Double Leg** | **Single Leg** | **Double Leg** | **Single Leg** |
| Hip | −69.9 ± 27.3 | −77.1 ± 27.6 | −67.8 ± 22.6 | * −81.9 ± 33.8 | −68.4 ± 21 | * −87.6 ± 50.7 |
| Knee | −36 ± 11.9 | −39.7 ± 19.2 | −36.1 ± 12.8 | −43.7 ± 27.7 | * −34.3 ± 10.7 | * −45.1 ± 30.6 |
| Ankle | 39.7 ± 6.2 | 41.2 ± 6.9 | 37.8 ± 6 | 38.4 ± 4.7 | * 23 ± 18.6 | * 16.7 ± 21.2 |

* Significant main effect at a level of less than 0.05. Negative directions indicate hip and knee flexion, positive directions indicates ankle dorsiflexion. LFL, leaning forward landing; SSL, self-selected landing; URL, upright landing.

## 4. Discussion

The current study hypothesized that, compared to the double-leg landing, single-leg landing would have a smaller hip and knee flexion angles at the time of initial foot contact with the ground, (2) have lower hip and knee flexion angular velocities at the time of initial foot contact with the ground, (3) have a greater GRF during landing, (4) have a greater knee joint reaction force during landing, and (5) have a greater knee flexion moment during landing. The present research has provided support through the obtained evidence

indicating significant levels for the torque angle during dorsiflexion, hip flexion and knee flexion ($p \leq 0.05$).

The contention that the shock absorption of the ankle joint angle is lessened in URL was further supported in our study ($16.7 \pm 21.2$). This observation indicates that LFL allowed the participants sufficient time to produce the peak ankle plantar flexor moment before the peak GRF occurred, increasing the proportion of GRF absorption by the ankle. Corresponding to requiring less dorsiflexion during landing (more plantar flexion), subjects showed significant results for the ankle (Table 2). Conversely, during URL, the peak GRF occurred before the peak ankle plantar flexor moment. Collectively, these results support previous suggestions that shock-attenuating capacity is higher in LFL than in URL [15]. Our results showed that the greatest peak knee extensor moment occurred during single-leg landing with the URL trunk position and the smallest peak knee extensor moment occurred during single-leg landing with the LFL trunk position. The hip extensor moments at the time of peak knee extensor moment were greater for URL than for LFL. Furthermore, the URL during single-leg landing is indicated as a position where the knee receives more external force and more extension; therefore, the chance of knee injury and consequently ACL injury will increase in sagittal plane trunk positions (Tables 2 and 3).

These findings indicate that trunk flexion potentially reduces a subsequent load placed on the ACL immediately after ground contact, which is when the ACL injury reportedly occurs. Trunk flexion during landing produced greater knee and hip flexion compared with a more erect or trunk-extended (URL) landing posture, placing the lower extremity in a position associated with decreased ACL injury risk [8]. As such, active trunk flexion could be an integral component of ACL injury-prevention programs by virtue of its ability to simultaneously influence kinetic, kinematic, and neuromuscular variables that have been suggested as risk factors for ACL injury.

## 5. Conclusions

This study suggests that leaning forward while landing (LFL) appears to protect the ACL by increasing the shock absorption capacity and knee flexion angles and decreasing anterior shear force due to the knee joint compression force. Conversely, upright landing (URL) appears to be harmful to the ACL by increasing the post-impact force of landing while decreasing knee flexion angles, all of which lead to a greater tibial anterior shear force and ACL loading. Based on the obtained data, we infer that a higher risk of ACL injury in the single-leg landing task could result from the fact that the single-leg landing exhibited greater peak proximal tibia anterior and lateral shear forces during the landing phase than the double-leg landed task (Tables 2 and 3). Further studies with a larger number of participants across sports (basketball, handball, soccer) would be required to provide new insights and support the data obtained in this present research.

Finally, authors acknowledge the limitation of the present study that is its small sample size. Although the results showed the LFL to be the best position for preventing the ACL injury, it remains to be proven, and without further research it would be an overstatement believing that lean forward landing could be an integral part of injury prevention programs. Moreover, how a forward lean could also impact the kinematics of landings in some sports (e.g., cause back injury, change dynamics of tactical components/movements, etc.) needs to be investigated.

**Author Contributions:** Methodology, M.K.; validation, T.Y., S.K. and S.T.; formal analysis, T.Y.; investigation, M.K.; data curation, T.Y.; writing—original draft preparation, M.K.; writing—review and editing, R.R.; supervision, R.R. All authors have read and agreed to the published version of the manuscript.

**Funding:** This research received no external funding.

**Informed Consent Statement:** Informed consent was obtained from all subjects involved in the study.

**Data Availability Statement:** The data are not publicly available due to ethical restrictions.

**Acknowledgments:** The authors would like to thank all the participants (women athletes) and research assistants who supported this research.

**Conflicts of Interest:** The authors declare no conflict of interest.

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
