# Peer review of "Correlation of Kinematics and Kinetics of Changing Sagittal Plane Body Position during Landing and the Risk of Non-Contact Anterior Cruciate Ligament Injury"

_applsci, doi:10.3390/app11177773_

Round 1

Reviewer 1 Report

Generally the paper has the potential to contribute to the area of injury risk reduction, however the authors need to substantially improve the readability of the paper which would include making a clearer and stronger link between what has been investigated and how this can impact lower-limb injuries (whether for better or worse). It is clear that the study has found a link between forward lean during landing, force dissipation and a potential for reduced injury, but the correlation appears weak and/or is not well discussed in the context of injury prevention for sport/athletes, generally or more specifically. Also, a discussion around the implications and practicality of encouraging a forward lean in landing techniques during sport should be provided, as this may potentially have follow on effects to other injuries, such as hamstring strain or back/head injury, depending on the sport.

The paper seems to be written in an unfinished format (track changes visible) and requires some major revisions to wording (English language) and clarity in some sections, including corrections to spelling and grammar. For example, there are two headings on page 4 of 9 (lines 137-139) that seem like information is missing or the headings be removed.

Methods:

The study methods require clearer explanation. Some areas to focus could include:

  • clearer definition of SSL - subjects were instructed to land any other way a part from URL or LFL, but what about if they felt comfortable landing upright, did this count as a successful trial?
  • what was the definition of a 'successful trial' as per line 106 and 127?
  • description of how the testing area (lines 92-94) was set-up was confusing even with a figure provided - needs clarification.
  • the description of the tests and test sequence (lines 108 to 116) of tests was disjointed and unclear - requires reword.

Results section:

  • remove 'N' column from table 3.1.1 and add '(n = 7)' in table heading.
  • Graph 1 (page 5 of 9) doesn't add value - suggest removing
  • remove wording in lines 198-199.
  • introduce Table 1 (text in lines 208-213) prior to presenting Table 1.
  • explanation of results in lines 214-215 is not correct according to table - knee single-leg landing during SSL for knee is not significant - correction to text or table required.

Conclusions:

  • The final conclusion (line 277-279) reads out of place and is unclear in the overall context of the paper - greater explanation required or remove.

Author Response

Dear Reviewer,

Thank you very much for your time and your comments. I would like to answer your comments by following:

Methods:

The study methods require a clearer explanation. Some areas to focus on could include:

  • clearer definition of SSL - subjects were instructed to land any other way apart from URL or LFL, but what about if they felt comfortable landing upright, did this count as a successful trial?

Answer: subject lands on the mid-foot with right trunk position

……………………………………………………………………………….

  • what was the definition of a 'successful trial' as per lines 106 and 127?

A: If participants did not place the dominant foot on the force plate, the lost balance did not execute the task with the appropriate landing technique or did not perform the appropriate task based on the cue generated by the software the trial was not successful and discarded from analysis

………………………………………………………………………………….

  • The description of how the testing area (lines 92-94) was set up was confusing even with a figure provided - needs clarification.

A: Clarified.

……………………………………………………………………………………. Results section:

  • remove 'N' column from table 3.1.1 and add '(n = 7)' in table heading.

Done

………………………………………………………………………….

  • Graph 1 (page 5 of 9) doesn't add value - suggest removing

Done

………………………………………………………………………….

  • remove wording in lines 198-199.

Done

………………………………………………………………………….

  • introduce Table 1 (text in lines 208-213) prior to presenting Table 1.

Done

………………………………………………………………………….

  • explanation of results in lines 214-215 is not correct according to the table - knee single-leg landing during SSL for the knee is not significant - correction to text or table required.

It has changed in the clear content.

………………………………………………………………………….

Conclusions:

  • The final conclusion (line 277-279) reads out of place and is unclear in the overall context of the paper - greater explanation required or remove.

Removed

………………………………………………………………………….

Reviewer 2 Report

Dear author,

the paper presents a poor description of the setup. In fact, it is mentioned markers were attached but no information is present about body placement. Then regarding the results and the conclusion, I see that you have conducted the experimentally in three subjects.
I suggest, before confirming your findings to extend this protocol to a broad population in order to respond also to the statistical evaluation that looks limited to only three subjects.

Overall, I want to give the opportunity to send the paper after this major revision.

Author Response

Dear Reviewer, Thank you very much for your time and your comments.

I did not bring the body placement in the paper because I thought would be unnecessary information. I have attached it here at the end and if you think would be mandatory to mention it I will add it in the methodology section. Regarding your other general comments, I made some changes and hopefully could convince you. please let me know if you have any questions.

''Forty-seven retro-reflective markers applied bilaterally on specific body landmarks over the right hand, right wrist lateral side, right wrist medial side, right elbow lateral side, right elbow medial side, right shoulder front side, right shoulder back side, right acromial, left hand, left wrist lateral side, left wrist medial side, left elbow lateral side, left elbow medial side, left shoulder front side, left shoulder back side, left acromial, right toe, right ball lateral side, right ball medial side, right heel, right ankle lateral side, right ankle medial side, right knee lateral side, right knee medial side, right trochanterion, left toe, left ball lateral side, left ball medial side, left heel, left ankle lateral side, left ankle medial side, left knee lateral side, left knee medial side, left trochanterion, top of head, right ear, left ear, suprasternal front, suprasternal back, right rib, left rib, xiphoid, xiphoid back, right anterior-superior iliac spine, left anterior-superior iliac spine, right posterior-superior iliac spine, left posterior-superior iliac spine.''

Round 2

Reviewer 1 Report

Some modifications are still required to strengthen the manuscript. Please review the attached PDF.

In addition, an acknowledgement of the study's limitations could be warranted given the small sample size. Saying that forward lean landing could be an 'integral part of injury prevention programs' is an overstatement without some acknowledgement of the study's capacity to fully prove this fact. This would also be the case without other discussion around how a forward lean could also impact the kinematics of landings in some sports (e.g. cause back injury, change dynamics of tactical components/movements, etc). It may be in the authors interests to add these sorts of comments.

Author Response

Dear Reviewer,

Thank you very much for your comments. I made changes as you requested. please find it in the word file.

Kind regards,

Mahgol

Reviewer 2 Report

Dear author, I suggest adding a more detailed description of the methods. As you replied marker placement or the protocol used is interesting for the readers. Line 17 of the abstract is mass or Kg? Please use the same reference system. Please re-check the entire manuscript.

Author Response

Dear Reviewer,

Thank you very much for your comments. I made changes as you requested. please find the attached file.

Kind regards,

Mahgol
